Purification and biochemical characterization of recombinant Persicaria minor β-sesquiphellandrene synthase

Ker De-Sheng 1
Pang Sze Lei 1
Othman Noor Farhan 1
Kumaran Sekar 1
Tan Ee Fun 1
Krishnan Thiba 2
Chan Kok Gan 2
Othman Roohaida 1 3
Hassan Maizom maizom@ukm.edu.my 1
Ng Chyan Leong clng@ukm.edu.my 1
1 Institute of Systems Biology, Universiti Kebangsaan Malaysia , Bangi , Selangor , Malaysia
2 Division of Genetics and Molecular Biology, Institute of Biological Sciences, Faculty of Science, University of Malaya , Kuala Lumpur , Malaysia
3 School of Biosciences and Biotechnology, Faculty of Science and Technology, Universiti Kebangsaan Malaysia , Bangi , Selangor , Malaysia
Silva Pedro
Electronic publication date: 2017 Feb 28
Publication date: 2017
Volume: 5
Electronic Location ID: e2961
Received 2016 Jul 19; Accepted 2017 Jan 5
Copyright: ©2017 Ker et al.
Copyright year: 2017
Copyright holder: Ker et al.
License: This is an open access article distributed under the terms of the Creative Commons Attribution License, which permits unrestricted use, distribution, reproduction and adaptation in any medium and for any purpose provided that it is properly attributed. For attribution, the original author(s), title, publication source (PeerJ) and either DOI or URL of the article must be cited.
License URL: https://creativecommons.org/licenses/by/4.0/

Keywords: Farnesyl diphosphate, β-sesquiphellandrene, Persicaria minor, Sesquiterpene synthase, Homology modelling

Funding: Ministry of Higher Education Malaysia (MOHE) Grant FRGS/1/2013/ST04/UKM/02/3 HIR Grants H-50001-A000027 A000001-50001 This work was supported by Ministry of Higher Education Malaysia (MOHE) Grant (FRGS/1/2013/ST04/UKM/02/3) and KGC was supported by the HIR Grants (H-50001- A000027 and A000001-50001). The funders had no role in study design, data collection and analysis, decision to publish, or preparation of the manuscript.

==============================
Background

Sesquiterpenes are 15-carbon terpenes synthesized by sesquiterpene synthases using farnesyl diphosphate (FPP) as a substrate. Recently, a sesquiterpene synthase gene that encodes a 65 kDa protein was isolated from the aromatic plant Persicaria minor. Here, we report the expression, purification and characterization of recombinant P. minor sesquiterpene synthase protein (PmSTS). Insights into the catalytic active site were further provided by structural analysis guided by multiple sequence alignment.

Methods

The enzyme was purified in two steps using affinity and size exclusion chromatography. Enzyme assays were performed using the malachite green assay and enzymatic product was identified using gas chromatography-mass spectrometry (GC-MS) analysis. Sequence analysis of PmSTS was performed using multiple sequence alignment (MSA) against plant sesquiterpene synthase sequences. The homology model of PmSTS was generated using I-TASSER server.

Results

Our findings suggest that the recombinant PmSTS is mainly expressed as inclusion bodies and soluble aggregate in the E. coli protein expression system. However, the addition of 15% (v/v) glycerol to the protein purification buffer and the removal of N-terminal 24 amino acids of PmSTS helped to produce homogenous recombinant protein. Enzyme assay showed that recombinant PmSTS is active and specific to the C15 substrate FPP. The optimal temperature and pH for the recombinant PmSTS are 30 °C and pH 8.0, respectively. The GC-MS analysis further showed that PmSTS produces β-sesquiphellandrene as a major product and β-farnesene as a minor product. MSA analysis revealed that PmSTS adopts a modified conserved metal binding motif (NSE/DTE motif). Structural analysis suggests that PmSTS may binds to its substrate similarly to other plant sesquiterpene synthases.

Discussion

The study has revealed that homogenous PmSTS protein can be obtained with the addition of glycerol in the protein buffer. The N-terminal truncation dramatically improved the homogeneity of PmSTS during protein purification, suggesting that the disordered N-terminal region may have caused the formation of soluble aggregate. We further show that the removal of the N-terminus disordered region of PmSTS does not affect the product specificity. The optimal temperature, optimal pH, Km and kcat values of PmSTS suggests that PmSTS shares similar enzyme characteristics with other plant sesquiterpene synthases. The discovery of an altered conserved metal binding motif in PmSTS through MSA analysis shows that the NSE/DTE motif commonly found in terpene synthases is able to accommodate certain level of plasticity to accept variant amino acids. Finally, the homology structure of PmSTS that allows good fitting of substrate analog into the catalytic active site suggests that PmSTS may adopt a sesquiterpene biosynthesis mechanism similar to other plant sesquiterpene synthases.

Introduction

Sesquiterpenes are a diverse group of 15 carbon long, volatile hydrocarbons assembled from three isoprenoid units, and are commonly found in plants, insects and fungi. Despite having only 15 carbon atoms, sesquiterpenes can be found forming many different and stereochemically complex structures in nature (Degenhardt, Köllner & Gershenzon, 2009). Utilizing farnesyl diphosphate (FPP), sesquiterpene synthases generate more than 200 different sesquiterpene hydrocarbon skeletons which serve as precursors for more than 7,000 derivative molecules (Cane, 1990; Misawa, 2011; Srivastava et al., 2015). The biosynthesis of sesquiterpenes is initiated by metal-dependent ionization of FPP, followed by a series of complex chemical mechanisms, involving isomerizations, cyclizations and rearrangements, catalyzed by sesquiterpene synthases, which then generate sesquiterpene products (Dickschat, 2011; Tantillo, 2011). Normally, each sesquiterpene synthase generates a single major sesquiterpene as its product; however, some sesquiterpene synthases are able to produce multiple different sesquiterpene products (Christianson, 2008; Degenhardt, Köllner & Gershenzon, 2009). For example, γ-humulene synthase from grand fir (Abies grandis) can produce 52 different sesquiterpenes (Steele et al., 1998). Nevertheless, the roles of majority of sesquiterpene synthases in guiding the specific mechanism of carbocation rearrangement to generate precise sesquiterpene remain unclear (O’Brien et al., 2016).

Taking advantage of the available transcriptome and genome data, functional genomics efforts have led to the discovery and characterization of sesquiterpene synthase genes from many fragrant plants including sweet wormwood (Artemisia annua) (Chang et al., 2000), tobacco (Nicotiana tabacum) (Back, Yin & Chappell, 1994), lavender (Lavandula angustifolia) (Landmann et al., 2007; Jullien et al., 2014) and sandalwood (Santalum album) (Jones et al., 2011; Srivastava et al., 2015). Persicaria minor is an aromatic plant widely distributed in Southeast Asia. It possesses a wide range of biological activities and is used locally as remedies for digestive disorder and dandruff (Christapher et al., 2014; Vikram et al., 2014). Previous chemical studies of P. minor have shown that P. minor essential oil contains mainly aldehydes and terpenes (Baharum et al., 2010; Ahmad et al., 2014), and sesquiterpenes are found predominantly in the flower (Prota et al., 2014). A few enzymes involved in flavonoid and terpenoid metabolite biosynthesis including geraniol dehydrogenase, chalcone synthase, and farnesol dehydrogenase have been identified in P. minor (Hassan et al., 2012; Roslan et al., 2012; Ahmad Sohdi et al., 2015).

Recently, a putative P. minor sesquiterpene synthase (PmSTS) gene (GenBank: JX025008) has been isolated. The PmSTS gene encodes a 562 amino acid protein and belongs to the TPS-a subfamily of angiosperm sesquiterpene synthases (Ee et al., 2014). The PmSTS gene has been cloned and expressed in Escherichia coli (Tan & Othman, 2012), gram positive bacteria Lactococcus lactis (Song et al., 2012) and in transgenic study of Arabidopsis thaliana (Ee et al., 2014). The His-tagged purified PmSTS from E. coli was found to produce α-farnesene (Tan & Othman, 2012), while His-tagged purified L. lactis recombinant PmSTSK266E (containing a K266E mutation introduced during cloning process) was reported to catalyze the formation of β-sesquiphellandrene (Song et al., 2012). Moreover, metabolite profile analysis of transgenic A. thaliana also indicated that PmSTS may be responsible for the formation of β-sesquiphellandrene. Note that none of these studies have purified the enzyme to homogeneity for enzyme characterization and activity assay. To clarify if PmSTS is an α-farnesene synthase or if the K266E mutation has changed the enzyme product to β-sesquiphellandrene, we conducted this work to purify the PmSTS to homogeneity for biochemical characterization. We report here the overexpression and purification of recombinant PmSTS protein in an E. coli system. The PmSTS was purified to homogeneity and used for enzyme characterization. The catalytic products were further analyzed using GC-MS. An homology model was utilized to provide insights into PmSTS active site in comparison with other sesquiterpene synthases.

Material and Methods

Materials

Pfu DNA polymerase was purchased from Biotechrabbit (Germany). Restriction enzymes and DNA ligase were purchased from Thermo Scientific (USA). DNA gel purification kits and plasmid purification kits were purchased from iNtRON Biotechnology (Korea). Farnesyl diphosphate (FPP), inorganic diphosphatase and standard alkane solution (C8–C20) were obtained from Sigma-Aldrich (St. Louis, MO, USA). Geranyl diphosphate (GPP) and geranylgeranyl diphosphate (GGPP) were purchased from Echelon Biosciences (Salt Lake City, UT, USA). Malachite Green Phosphate Assay kit was obtained from Bioassay Systems (Hayward, CA, USA). QuikChange site-directed mutagenesis kit was obtained from Agilent Technologies (Santa Clara, CA, USA). HisTrap™ HP 5 mL, and HiLoad 16/600 Superdex 200 pg were purchased from GE Healthcare (Chicago, IL, USA).

Design of recombinant P. minor sesquiterpene synthase constructs (PmSTS-Δ24)

To remove the N-terminal disordered region and enhance protein homogeneity, a truncated PmSTS construct, namely PmSTS_Δ24 was generated from the full-length recombinant PmSTS using forward primer (5′-GCCCCTCGTCATATGGCAGGTTTCAAACCTTCC-3′) and reverse primer (5′-CCAAGCTTTCATATCAGTATGGGATCGATGTAC-3′). The NdeI and HindIII restriction endonuclease sequences are underlined in these oligonucleotides, and the stop codon UGA is indicated in bold characters. The PCR amplification was performed according to the manufacturer’s guidelines. The PCR product was analyzed by agarose gel electrophoresis and purified using DNA gel purification kit following manufacturer’s guidelines. The purified PCR product was digested with NdeI and HindIII and ligated into pET28b expression vector.

Molecular cloning

The full length PmSTS (GenBank accession no: JX025008), and truncated PmSTS_Δ24 were cloned into pET28b with the affinity tag (His6) at its N-terminus, which yielded the resulting recombinants, pET28b_PmSTS and pET28b_PmSTS_Δ24, respectively. The plasmids were transformed into competent E. coli TOP 10 cells using heat shock at 42 °C for 60 s and the transformants were selected on LB plates containing kanamycin (50 µg mL−1). Positive colonies were identified by colony PCR. Recombinant plasmid was isolated from positive transformants using a plasmid purification kit. The constructs were verified by Sanger DNA sequencing (First BASE Laboratories Sdn Bhd, Malaysia).

Protein expression of recombinant P. minor sesquiterpene synthase (PmSTS)

The recombinant plasmids of pET28b_PmSTS and pET28b_PmSTS_Δ24 were transformed into competent E. coli BL21 (DE3) cells. The effect of different temperature on the solubility of recombinant protein expression was investigated by isopropyl β-D-1-thiogalactopyranoside (IPTG) induction at 37 °C and 16 °C. Briefly, a single colony was picked and cultured overnight at 37 °C in 10 mL of sterile LB culture (50 µg mL−1 of kanamycin) with agitation 200 rev min−1. The cells were allowed to grow at 37 °C until OD600 reached 0.6. The cultures grown at 37 °C and 16 °C were induced by adding IPTG to a final concentration of 0.5 mM. The culture was further incubated at 37 °C and 16 °C for 4 h and 16 h at 200 rev min−1 respectively. Cells were harvested by centrifugation for 10 min at 5,500 g at 4 °C and stored frozen at −80 °C until use.

Cells that had been induced at 37 °C and 16 °C were resuspended in lysis buffer containing 20 mM Tris–HCl (pH 8.0), 500 mM NaCl, 20 mM β-mercaptoethanol (βME), and lysed with 10 min sonication composed of 5 s pulse with 10 s rest at amplitude 30% power using ultrasonicator (QSONICA) on ice. Cell lysate was then centrifuged at 13,000 g for 30 min at 4 °C. The pellet and supernatant corresponding to insoluble and soluble proteins were analyzed using SDS-PAGE.

Purification of recombinant P. minor sesquiterpene synthase (PmSTS)

For protein purification, cell pellets harvested from 2 L of LB culture were used. The cell pellet was resuspended in 30 mL of binding buffer (20 mM Tris–HCl, pH 8.0, 500 mM NaCl, 20 mM βME, 20 mM imidazole). For purification in the presence of glycerol, binding buffer G (20 mM Tris–HCl, pH 8.0, 500 mM NaCl, 20 mM βME, 20 mM imidazole, 15% (v/v) glycerol) was used instead. The cell suspensions were disrupted with a 10 min sonication composed of 15 s pulse with 30 s rest at amplitude of 30% power using an ultrasonicator (QSONICA) on ice. Cell lysate was then centrifuged at 13,000 g for 30 min at 4 °C. The supernatant fraction was filtered through a syringe filter (0.2 µm pore size) before being applied into a HisTrap™ HP 5 mL (GE Healthcare), pre-equilibrated with binding buffer. After washing of HisTrap™ with 10 column volumes (CV) of binding buffer, protein was eluted with elution buffer (20 mM Tris–HCl, pH 8.0, 500 mM NaCl, 20 mM βME, 500 mM imidazole) in 2 mL fractions by 20 CV in linear gradient. Eluted protein fractions were pooled and concentrated to 2 mL using Microsap Advanced Centrifugal Device (10 kDa MWCO; Pall, New York, NY, USA) at 4 °C and were further purified using HiLoad 16/600 Superdex 200pg (GE Healthcare) at a flow rate of 0.8 mL min−1. Eluted protein fractions were pooled and concentrated using a Microsap Advanced Centrifugal Device (10 kDA MWCO; Pall, New York, NY, USA).

Enzyme assay

Enzyme assays were performed using Malachite Green Phosphate Assay Kits (BioAssay Systems) in 96-well flat bottom plates. After purification, protein concentrations were determined using the Bradford Assay (Amesco). Briefly, 0.1 µM of purified enzyme was equilibrated in reaction mixture containing 20 mM HEPES, pH 8.0, 10 mM MgCl2, 100 mU inorganic diphosphatase for 2 min at room temperature. Reactions (240 µL) were started by the addition of FPP, GPP or GGPP, and allowed to proceed at 30 °C for 5 min. After incubation, 80 µL of reaction mixture were transferred to each well (96-well plate) and the enzyme reactions were quenched by addition of 20 µL of malachite green solution. After 20 min of incubation, reactions were read at 655 nm using an iMark plate reader (Bio Rad). Negative controls were performed without the addition of purified enzyme. Monophosphate (Pi) and diphosphate (PPi) were generated according to the instructions of the manufacturer.

Enzyme characterization

The optimal temperature was determined in a series of temperatures ranging from 25 °C to 55 °C in 20 mM HEPES buffer (pH 8.0). The optimal pH was determined at room temperature from pH 6.0 to pH 10.5 using 20 mM BIS-TRIS propane, and 20 mM glycine NaOH buffer. Kinetic parameters were determined in assays with ten different substrate concentrations (2–30 µM) at pH 8.0 and 30 °C using 0.1 µM of purified enzyme. Apparent Km, kcat and kcat∕Km values were obtained with GraphPad Prism 5 software.

Product identification using gas chromatography mass spectrometry (GC-MS)

For the product identification of PmSTS and PmSTS_Δ24, two extraction methods were used: headspace solid phase microextraction (HS-SPME) and solvent extraction.

For the HS-SPME extraction method, PmSTS or PmSTS_Δ24 (∼80 µg) was incubated with substrate (60 µM FPP) in assay buffer (500 µL) containing 20 mM HEPES (pH 8.0), 10 mM MgCl2, and 1 mM dithiothreitol. The reaction mixture was then incubated at 30 °C for 2 h, and the reaction products were extracted by HS-SPME using 100 µm polydimethylsiloxane coated fiber (Supelco, Bellefonte, PA, USA). Headspace sampling times using SPME was 30 min at 45 °C and the products were analyzed using GC-MS. GC-MS analysis was performed as described previously (Song et al., 2012; Tan & Othman, 2012) using Perkin Elmer, Turbomass Clarus 600 equipped with Perkin Elmer Elite 5 MS (30 m length, I.D. 0.25 mm, 0.25 µm film thickness).

For the solvent extraction method, the reaction mixtures were overlaid with 200 µL of hexane to trap the reaction product. The PmSTS, PmSTS_Δ24 and PmSTS_Δ24_K266E protein were incubated overnight. After incubation, the hexane layer was extracted and subjected to GC-MS analysis. GC-MS analysis was performed as described previously (O’Maille, Chappell & Noel, 2004) on an Agilent 7890A gas chromatograph equipped with HP-5MS (30 m length, I.D. 0.25 mm, 0.25 µm film thickness) and 5975C MSD with triple-axis detector.

In both methods, products were identified based on their mass spectra and Kovats Index, calculated in relation to the retention times of a series of alkanes (C8–C20). The mass spectra were compared to those in the National Institute of Standards and Technology (NIST) Library in 2011.

Protein disordered region predictions

The following servers were used for disordered region prediction of PmSTS: DISOPRED3 (Jones & Cozzetto, 2015), DisEMBL (Linding et al., 2003), and RONN (Yang et al., 2005). In all cases, PmSTS (residue 1–562) was subjected to disorder prediction using default server parameters.

Multiple sequence alignment and homology modelling

Sequences of plant sesquiterpene synthase were obtained from the SWISS-PROT database through a text search for sesquiterpene. Protein sequences of 500–600 residues were retained and the proteins that produce sesquiterpene, as judged from available GC-MS analysis, were selected. Multiple sequence alignment (MSA) was performed using Clustal Omega webserver (Sievers et al., 2011). The alignment was then visualized and analyzed using Jalview 2 (Waterhouse et al., 2009). An homology model of PmSTS as previously reported (Ee et al., 2014) was constructed using I-Tasser (Roy, Kucukural & Zhang, 2010). Superimposition of PmSTS with other terpene synthases (PDB: 3M01, 5EAT, 4FJQ, 3G4D, 1N20, 2ONG) were performed using Pymol (The PyMOL Molecular Graphics System, Version 1.8 Schrödinger, LLC.).

Site-directed mutagenesis to generate PmSTS_Δ24K266E mutant

To generate the K266E mutant, site-directed mutagenesis was performed using QuikChange site-directed mutagenesis kit (Agilent) according to the manufacturer’s guidelines. The PCR-based mutagenesis protocol was performed with the PmSTS_Δ24 cDNAs cloned into the expression vector pET28b using forward primer (5′-GAAATGTGCAGGTGGTGGG AAAAGGTGAATATGACTAAG-3′) and reverse primer (5′-CTTAGTCATATTCACCTTTT CCCACCACCTGCACATTTC-3′). The mutagenized construct was fully sequenced before expression. The overexpression and purification of PmSTS_Δ24K266E mutant protein was performed the same for the recombinant PmSTS.

Results and Discussion

Protein disordered region analysis of PmSTS lead to construct design of PmSTS_Δ 24 protein

Analysis of PmSTS sequence using several disorder prediction servers suggested that the N-terminal region of PmSTS (about 1–25 amino acids residues) contains disordered regions (Fig. S1). The disordered regions are amino acid regions that lack a stable secondary structures and have high conformational dynamics and flexibility that are susceptible to aggregation (Lebendiker & Danieli, 2014). To eliminate the possibility of protein aggregation caused by the disordered region, and hence facilitate the purification of homogenous protein, the recombinant PmSTS_Δ24 was designed by truncating the N-terminal predicted disordered region.

Cloning and over expression of recombinant P. minus sesquiterpene synthase (PmSTS)

Full length recombinant sesquiterpene synthase of P. minor (PmSTS) and N-terminally truncated variant (PmSTS_Δ24) were overexpressed with pET28b vector using E. coli BL21 (DE3) strain. Both the PmSTS and the PmSTS_Δ24 recombinant proteins contain a His6-tag at its N-terminus to aid in the purification of recombinant protein using immobilized metal affinity chromatography (IMAC). To monitor the expression level and solubility properties of PmSTS proteins, E. coli harboring the PmSTS or PmSTS_Δ24 gene was expressed at two different temperatures, 16 °C and 37 °C. Recombinant cells grown at both 37 °C and 16 °C showed the production of recombinant enzyme, however soluble protein expressions of PmSTS and PmSTS_Δ24 were only observed at 16 °C (Fig. 1). Expression at 37 °C drove all the proteins into inclusion bodies (Fig. 1). Lower growth temperature is known to facilitate the production of soluble recombinant protein through slowing down the transcription and translation rates, as well as reducing the strength of hydrophobic interactions that contribute to protein misfolding (Baneyx & Mujacic, 2004). The truncated version of PmSTS (PmSTS_Δ24) did not show an enhanced protein expression solubility despite the removal of the N-terminal disordered region (Fig. 1B).

Figure 1 Expression analysis of (A) recombinant PmSTS and (B) truncated recombinant PmSTS_Δ24 from E. coli BL21 (DE3).

Soluble protein expression of PmSTS and PmSTS_Δ24 were only observed at growth temperature 16 °C. M, protein marker (kDa). 1, Uninduced sample. 2, Sample induced with 0.5 mM IPTG at 37 °C. 3, Sample induced with 0.5 mM IPTG at 16 °C.

Figure 2 Size exclusion chromatography (SEC) and SDS-PAGE profile of PmSTS.

(A) SEC and SDS PAGE profile of PmSTS without presence of glycerol in the purification buffer. (B) SEC and SDS PAGE profile of PmSTS in the presence of 15% (v/v) glycerol in the purification buffer. (C) SEC and SDS PAGE profile of truncated PmSTS_Δ24 in the presence of 15% (v/v) glycerol in the purification buffer. M, protein molecular weight markers (kDa); 1, Soluble fraction of uninduced cell lysate; 2, Soluble fraction of induced cell lysate; 3, Protein purified by immobilized metal affinity chromatography (IMAC); 4, Protein fraction from peak 4 in SEC; 5, Protein fraction from peak 5 in SEC.

Purification of recombinant PmSTS enzyme

The recombinant protein purification was conducted using nickel affinity chromatography. The recombinant protein was eluted at 200 mM imidazole, and the eluted protein fractions were identified by SDS-PAGE. The results showed high levels of E. coli contaminants eluted together with PmSTS (Fig. 2A, lane 3). Additional protein purification using size exclusion chromatography (SEC) indicated that PmSTS may have bound together with the contaminants and formed soluble aggregates (Fig. 2A). Further protein purification buffer optimization has identified that addition of 15% (v/v) glycerol in protein purification buffer has aided in reducing the contaminants and improved the homogeneity of the PmSTS protein (Fig. 2B), although the majority of the protein still remained as soluble aggregates. The SEC profile suggested that PmSTS exists as a monomer in solution. The PmSTS_Δ24 was expressed in E. coli BL21 (DE3) and purified using identical method as for PmSTS. The results revealed significant improvement in the purification profile of homogeneous PmSTS_Δ24 compared to PmSTS (Fig. 2C). Both systematic protein purification optimization and truncated protein design suggested that addition of glycerol in the buffer and elimination of N-terminus disordered region of PmSTS are important in producing homogenous PmSTS enzyme.

Enzyme characterization of PmSTS

The purified full length PmSTS and PmSTS_Δ24 proteins were used for biochemical characterization. Some sesquiterpene synthases have been reported to have broad substrate specificity, accepting both GPP and FPP as substrates (Nieuwenhuizen, Wang & Matich, 2009; Zhuang et al., 2012; Srivastava et al., 2015). However, the enzyme activity assay showed that both PmSTS and PmSTS_Δ24 were only active towards C15 substrate farnesyl diphosphate (FPP). Neither C10 substrate geranyl diphosphate (GPP) nor C20 substrate geranylgeranyl diphosphate (GGPP) are substrates for PmSTS and PmSTS_Δ24.

As for the determination of kinetic parameters, only the activity of PmSTS_Δ24 was assayed. The purified full length PmSTS displayed enzyme instability and losing its activity over a period of time, thus making it unsuitable for the enzyme assay. The cause of the instability of PmSTS protein is yet to be investigated.

The optimal temperature for PmSTS_Δ24 activity was found at 30 °C (Fig. 3A). At 40 °C, the enzymatic activity was less than 40% compared to that at 30 °C. The optimal pH range of the enzyme was further determined at pH 7.5 to 8.0 (Fig. 3B). The enzyme activity was found to be dramatically reduced above pH 8.5. Kinetic characterization of PmSTS_Δ24 on FPP was also performed. PmSTS_Δ24 has an apparent Km value of 10.2 µM, a kcatvalue of 0.078 s−1 and kcat∕Km of 7.6 × 103 M−1 s−1. The optimal pH and temperature, as well as the kinetic parameters of PmSTS_Δ24 are comparable to other plant sesquiterpene synthases (Table 1). As judged by the kcat∕Km values, the plant sesquiterpene synthases, in general, have low catalytic activity. The low catalytic activity is common as the plant sesquiterpene synthases are plant enzymes involved in secondary metabolism, which are known to have kcat∕Km values around 103 M−1 s−1 or lower (Bar-Even et al., 2011).

Figure 3 Biochemical analysis of PmSTS_Δ24.

The purified PmSTS_Δ24 was incubated with farnesyl diphosphate (FPP) at (A) different temperature and (B) different pH. (C) Michaelis–Menten plot for PmSTS_Δ24. Error bars denote standard deviation (n = 3). Error bars denote standard deviation (n = 3).

Table 1 Biochemical data of PmSTS compared with other plant sesquiterpene synthases.

	PmSTS_Δ24	1	2	3	4	
pH	8.0	7.5	8.0	6.5	7.7	
Temperature	30 °C	40 °C	30 °C	–	–	
Km (µM)	10.2	4.45	4.7	2.1	1.8	
kcat (s−1)	7.8 × 10−2	4.3 × 10−4	3.3 × 10−2	9.5 × 10−3	4.0 × 10−2	
kcat∕Km (M−1 s−1)	7.6 × 103	96.6	7.0 × 103	4.5 ×  103	2.2 ×  104	
Notes.

1 Patchoulol synthase from Pogostemon cablin (Deguerry et al., 2006)

2 α-Bergamotene synthase from Lavandula angustifolia (Landmann et al., 2007)

3 β-Farnesene synthase from Artemisia annua (Picaud, Brodelius & Brodelius, 2005)

4 β-Caryophyllene synthase from Artemisia annua (Cai et al., 2002)

P. minor sesquiterpene synthase produce β-sesquiphellandrene

The enzyme assays of PmSTS and PmSTS_Δ24 with FPP as substrate were performed by headspace solid phase microextraction gas chromatography mass spectrometry (HS-SPME-GC-MS). The HS-SPME-GC-MS analysis of volatile sesquiterpene produced by PmSTS and PmSTS_Δ24 identified β-sesquiphellandrene as the main product (∼97%) and β-farnesene (∼3%) as a minor product (Fig. 4). The HS-SPME result was further verified by solvent extraction using hexane, which showed that the major product is indeed β-sesquiphellandrene (KI:1516).

Figure 4 Headspace solid phase microextraction gas chromatography mass spectrometry (HS-SPME-GC-MS) analysis of enzymatic products of recombinant PmSTS and truncated recombinant PmSTS_Δ24.

(A) GC-MS chromatogram of sample extracted from in vitro enzymatic reaction containing PmSTS. (B) GC-MS chromatogram of sample extracted from in vitro enzymatic reaction containing PmSTS_Δ24. (C–D) GC-MS mass spectra for the compounds of 1 and 2 in (A) and (B). According to the NIST11 mass spectral library, the compound 1 and 2 were identified as β-farnesene and β-sesquiphellandrene, respectively.

Previous GC-MS analysis of enzymatic reaction using partially purified recombinant PmSTS from E. coli showed that PmSTS produced α-farnesene (Tan & Othman, 2012). However, further studies using partially purified recombinant PmSTS from L. lactis (Song et al., 2012) and metabolite studies of A. thaliana expressing PmSTS (Ee et al., 2014), have shown that PmSTS is a β-sesquiphellandrene synthase. In this study, the PmSTS that had been purified to homogeneity was further confirmed as a β-sesquiphellandrene synthase. In an effort to prove that the point mutation K266E, introduced during the cloning process in L. lactis (Song et al., 2012), does not interfere in product specificity of PmSTS, site-directed mutagenesis was undertaken to alter residue lysine 266 to glutamic acid. The recombinant mutant protein K266E produced using an E. coli expression system displayed a product profile resembling that of PmSTS (Fig. S5). Structural analysis revealed that residue K266E is located at the exterior surface of PmSTS (Fig. 5C), and therefore is unlikely to affect the PmSTS product specificity. The truncated PmSTS with the removal of N-terminal 24 residues (PmSTS_Δ24) was shown to synthesize β-sesquiphellandrene as a major product, identical to the full length PmSTS. Thus, the N-terminal region is also not involved directly in product specificity of PmSTS, and similar properties have also been reported for truncated γ-humulene synthase from grand fir (Little & Croteau, 2002).

β-Sesquiphellandrene is a sesquiterpene found as a constituent of ginger (Zingiber officinale) (Onyenekwe & Hashimoto, 1999), turmeric (Curcuma longa) (Tyagi et al., 2015), and Alpinia conchigera (Ibrahim et al., 2009). Previous studies have shown that β-sesquiphellandrene exhibits various biological activities such as antioxidant and anticancer activities (Zhao et al., 2010; Tyagi et al., 2015). Besides plants, β-sesquiphellandrene has been found in insect as sex pheromone (Borges et al., 2007). In P. minor, sesquiphellandrene was detected at minute amount (0.1%) in the leaves and stems (Ahmad et al., 2014).

Some plant sesquiterpene gene expressions have known to be mediated by plant developmental stages or environmental stresses (Bohlmann et al., 1998; Xu et al., 2004; Zhuang et al., 2012; Yu et al., 2015). For example, sesquiterpene synthases of rice (Os08g07100) and sorghum (SbTPS1, SbTPS2), that share 32%–35% sequence identity with PmSTS, were found to produce β-sesquiphellandrene after damage by herbivores, suggesting that the emission of β-sesquiphellandrene play a role in crop defense (Yuan et al., 2008; Zhuang et al., 2012). The low abundance of sesquiphellandrene detected in P. minor may indicate similar regulation in the expression of sesquiterpene synthase (PmSTS).

Multiple sequence alignment of plant sesquiterpene synthases and PmSTS reveals an altered second metal binding motif

BLAST analysis revealed that PmSTS has high homology with sesquiterpene synthases of angiosperms, with the highest level of similarity (45%) to drimenol synthase, a cyclic sesquiterpene synthase, from Persicaria hydropiper (GenBank Accession No: KC754968.1). PmSTS contains numerous motifs highly conserved among the plant sesquiterpene synthases, including RXR, DDXXD and NSE/DTE motifs (Ee et al., 2014). However, extensive MSA analysis in this study unexpectedly discovered that PmSTS contains a modified metal binding motif N458DXXG462XXXV466 on helix H. This metal binding motif usually has consensus sequence (N/D)DXX(T/S/G)XXXE (Christianson, 2006; Zhou & Peters, 2009) or DDXX(D/E) (Gennadios et al., 2009) in which boldface residues typically binds to one Mg cofactor, namely Mg2+B.

Figure 5 The homology model of PmSTS shows the structure domain and active site of the enzyme.

(A) The enzyme is made up of α-helices architecture structure to contain terpene synthase family N-terminal domain (blue) and C-terminal metal-binding domain (green). The truncated N-terminal (24 amino acid residues) disordered region of PmSTS_Δ24 is colored in orange. (B) Superimpose of PmSTS (green) to monoterpene synthase S. officinalis (+)-bornyl diphosphate synthase (SoBDS) (PDB ID:1N20 in purple) and M. spicata 4S-limonene synthase (MsLS) (PDB ID:2ONG in yellow), and sesquiterpene N. tabacum 5-epi-aristolochene synthase (NtEAS) (PDB:3M01 in brown), G. arboreum δ-cadinene synthase(GaDCS) (PDB ID: 3G4F in cyan) and A. annua α-bisabolol synthase (AaBOS) (PDB ID: 4FJQ in black), reveals the structural conserved RXR and DDXXD motifs, and flexible region (boxed) of J–K and H2-α1 loops at the active site. The ligand FPF and trinuclear metal cluster were adopted by superimposed NtEAS structure with PmSTS (C) The active site of PmSTS shows the ligand entrance pocket and the potential enzyme-substrate interactions. The three catalytic important Mg2+ ions are also shown in magenta sphere. The mutated residues K266E that found in L. lactis recombinant protein PmSTS K266E is as stick in helix A.

Figure 6 Multiple sequence alignment of sesquiterpene synthase metal binding conserved motifs for selected plant sesquiterpene synthases.

The first metal binding motif is highly conserved among the plant sesquiterpene synthases and has a consensus sequence of DDXXD. The second metal binding motif is less conserved and has a consensus sequence of either DDXX(D/E) or (N/D)DXX(T/S/G)XXXE. In PmSTS, the second metal binding motif has the (N/D)DXX(T/S/G)XXXE consensus sequence with alteration, where the conserved E residue is replaced by V466 as denoted by asterisk. AtGBS (GenBank: CP002687); GhDCS (GenBank: U88318); AgGHS (GenBank: U92267); PaLS (GenBank: AY473625); GaDCS (GenBank: U23206); MdAFS (GenBank: AY182241); HaGCS (GenBank: DQ016668); SbBSS (UniProt: C5YHI2); NtEAS (GenBank: L04680), ZoBBS (GenBank: AB511914); LaBCS (GenBank: DQ263742).

In PmSTS, the glutamate residue in metal binding motif (NDXXGXXXE) is found to be replaced by valine residue, N458DXXG462XXXV466, indicating that PmSTS contains a modified metal binding motif (Fig. 6). This alternative form of motif has not been reported in plant sesquiterpene synthases. Previous mutational analyses on other terpene synthases have shown that changes from glutamate to glutamine or aspartate in the metal binding motif greatly reduce the catalytic activity and changes the product specificity of terpene synthases (Peters & Croteau, 2002; Felicetti & Cane, 2004). However, PmSTS was found to be fully active, despite the fact that the hydrophobic side chain of V466 is not able to form a hydrogen bond with a Mg2+ ion. It is likely that the side chain of N458 may still chelate the Mg2+B ion, with assistance from a water molecule (Zhou & Peters, 2009) or carboxylic side chain of D459 (Fig. S7), thereby PmSTS has a catalytic efficiency (kcat∕Km) that is comparable to other sesquiterpene synthases (Table 1).

Multiple sequence alignment between linear and cyclic plant sesquiterpene synthase was performed to identify a potential conserved motif responsible for the cyclization in sesquiterpene synthases. However, MSA analysis did not find any motif that was able to distinguish between linear and cyclic sesquiterpene synthases. A similar result was obtained from phylogenetic analysis of various α-farnesene synthases with other terpene synthases, as the phylogenetic analysis did not cluster all α-farnesene synthase together in one group (Green et al., 2007). This is not surprising given the low sequence identity within sesquiterpene synthase family (Christianson, 2006; Aaron & Christianson, 2010), even though most of the enzymes share a similar overall structure. Furthermore, based on current knowledge of sesquiterpene biosynthesis, it is still not possible to predict the absolute sesquiterpene product of a sesquiterpene synthase based on its amino acid sequence (Zulak & Bohlmann, 2010; Dickschat, 2011).

Homology model provides structural insights into PmSTS catalytic active site

The previously generated homology model was used to gain structural insight into PmSTS catalytic mechanism (Ee et al., 2014). The overall structure of PmSTS adopts an α-helical architecture containing two domains that resemble the terpene synthase family N-terminal domain (residue 1–236 for PmSTS) and terpene synthase family C-terminal metal-binding domain (residue 237–562 for PmSTS), also known as a catalytic domain (Fig. 5A). Based on the homology model, the predicted N-terminal 24 amino acid disordered region is positioned near the entrance of the active site; however, it does not affect the enzyme product specificity. Overall, the PmSTS structure is highly similar to other plant terpene synthases (Table 2). Despite lower sequence identity, PmSTS structure shared more similarity to the monoterpene synthase structure with lower RMSD than to its sesquiterpene synthase counterpart (Table 2). Structural comparison further revealed that the J–K loop and the second metal-binding motif region (H2 helix and H2-α1 loop) are highly flexible compared to the conserved motif RXR and DDXXD located at helix D of the active site (Fig. 5B).

Table 2 Structural similarity of between PmSTS and other terpene synthases.

	Sesquiterpene	Monoterpene	
	NtEAS	AaBOS	GaDCS	SoBDS	MsLS	
PDB accession code	3M01	4FJQ	3G4D	1N20	2ONG	
Organism	Nicotiana tabacum	Artemisia annua	Gossypium arboreum	Salvia officinalis	Mentha spicata	
Terpene synthase	5-Epi- aristolochene	α-Bisabolol synthase	δ-Cadinene synthase	Bornyl diphosphate synthase	Limonene synthase	
Sequence identitya	37.6%	37.0%	40.0%	29.5%	28.8%	
RMSDb	1.43 (518)	2.26 (486)	1.53 (502)	1.15 (511)	1.26 (517)	
Notes.

a Sequence identity compared with PmSTS.

b The rms deviation of Cα atoms and the number of structurally similar residues (in parentheses) compared with PmSTS.

Superimposition of the PmSTS homology model to the substrate analog complex of N. tabacum 5-epi-aristolochene synthase (NtEAS; PDB:3M01) and Gossypium arboreum (+)-delta-cadinene synthase (GaDCS; PDB:3G4F) showed that PmSTS may adopt a substrate binding like NtEAS (Fig. 5B), but is unable to bind ligand as seen in GaDCS due to the ligand binding mode of GaDCS, which may cause steric clash with the J–K loop of PmSTS (Fig. S8).

To better elucidate the function of β-sesquiphellandrene synthase, a structural study of PmSTS in complex with an FPP analog will be important to provide insights into the active site especially at the modified second metal-binding motif that is likely to interact with the Mg2+B ion.

Based on previous knowledge about the reaction mechanism of other sesquiterpene synthases (Köllner, Gershenzon & Degenhardt, 2009; McAndrew et al., 2011; Garms et al., 2012), we proposed a reaction mechanism for PmSTS such that the biosynthesis of sesquiterpene begins with metal-dependent ionization of the diphosphate moiety of FPP to form a farnesyl cation and a diphosphate group (Fig. 7). The diphosphate group will interact with and be stabilized by three Mg2+ ions and highly conserved positively charged residues, R277, R279, and R455. The positively charged region will direct the diphosphate away from the active site. Deprotonation of the farnesyl cation may yield the minor product β-farnesene. However, most of the farnesyl cation will undergo 1,6 cyclization via a nucleophilic attack of the C6–C7 double bond generating a bisabolyl cation. Subsequently, the 1,3-hydride shift between C1 and C7 and deprotonation at C15 will lead to the formation of β-sesquiphellandrene (Garms et al., 2012).

Figure 7 Proposed formation of the two sesquiterpene products from FPP catalyzed by PmSTS.

The scheme is based on previous study on β-sesquiphellandrene synthase from Sorghum bicolor (Garms et al., 2012).

Conclusion

The sesquiterpene synthase gene from P. minor was successfully cloned, expressed and purified to homogeneity for the first time using an E. coli expression system. The truncation of the predicted unstructured N-terminal region of PmSTS dramatically increased the homogeneity of PmSTS, thus indicating that N-terminal disordered region may be one of the causes of PmSTS protein aggregation. The combination of 15% (v/v) glycerol in protein purification buffer and elimination of the N-terminal disorder region of PmSTS are particularly important to produce homogenous PmSTS enzyme. These findings serve as an important example for the production of homogenous recombinant plant sesquiterpene synthases and may provide valuable information for future structural studies of PmSTS. GC-MS analysis revealed that both the full length PmSTS and truncated PmSTS_Δ24 recombinant proteins are active and produce mainly β-sesquiphellandrene.

Biochemical characterization of PmSTS showed that PmSTS utilizes FPP as a substrate and shares typical plant sesquiterpene synthases characteristics. No enzyme activity was detected when GPP or GGPP were used as substrate. Sequence alignment analysis identified a previously unreported altered conserved metal binding motif N458DXXG462XXXV466 in PmSTS, suggesting that sesquiterpene synthases are able to accommodate variant amino acid at this location. Finally, homology modelling and structural analyses suggest that PmSTS may likely bind to substrate in a similar manner as to tobacco 5-epi-aristolochene synthase.

Supplemental Information

Figure S1 Disorder region predictions of PmSTS (1–562)

Black- residues predicted to be disordered. Each row represents the results from the disorder prediction server of DISOPRED (Jones & Cozzetto, 2015), RONN (Yang et al., 2005) and DisEMBL (Linding et al., 2003), respectively.

Click here for additional data file.

Figure S2 Calibration curve on HiLoad Superdex 200 pg and molecular weight of PmSTS

The size exclusion chromatography column was calibrated with the following protein standards: thyroglobulin (670 kDa), γ -globulin (158 kDa), ovalbumin (44 kDa), and myoglobin (17 kDa). The elution pattern of the protein size markers was linear on a semilog plot. Elution data are represented as log molecular weight to Kav. Kav was calculated as in the equation (Ve − Vo)/(Vt − V0), Ve, Elution volume; Vo, Void volume (determined by the elution of Blue dextran, 2,000 kDa); Vt, total column volume. Calculated and measured molecular weight values of His-tagged PmSTS and His-tagged PmSTS_Δ24.

Click here for additional data file.

Figure S3 The SEC profile of PmSTS shows the effect of glycerol concentration in the purification buffer on homogeneity of PmSTS

For comparison purpose, all SEC chromatogram is normalized based on the height of peak (Aggregate).

Click here for additional data file.

Figure S4 Time course of enzyme activity for PmSTS_Δ 24

Click here for additional data file.

Figure S5 (A) The GC-MS profile of PmSTS_Δ 24_K266E and PmSTS_Δ24. (B) Mass spectra of the major peak (17.493 min) in PmSTS_Δ 24_K266E

Click here for additional data file.

Figure S6 Comparison of amino acid sequences of PmSTS with four plant sesquiterpene synthases

The alignment was generated by T-Coffee and drawn with ESPript. Consensus amino acid residue are boxed in black, secondary structure elements of NtEAS (PDB: 5EAS) are shown above the sequences. The conserved motifs (RXR, DDXXD, and NSE/DTE) are underlined in black. Naming of helices was based on the convention used for NtEAS (Starks et al., 1997). The sesquiterpene synthase sequences aligned are NtEAS (N. tabacum 5-epi-aristolochene synthase), GaDCS (G. arboreum δ -cadinene synthase), AaBOS (A. annua α-bisabolol synthase), MaLS (M. spicata limonene synthase) and SoBDS (S. ofiicinalis bornyl diphosphate synthase).

Click here for additional data file.

Figure S7 The NSE/DTE motif in sesquiterpene synthase active site

Superimpose of PmSTS (Green) to N. tabacum 5-epi-aristolochene synthase (NtEAS) (PDB:5EAT in brown). The Mg 2+B is shown as magenta sphere. Oxygen and nitrogen atoms are coloured red and blue, respectively. Important distances and likely hydrogen bonds are shown by dashed lines.

Click here for additional data file.

Figure S8 Superimposition shows steric clash between J-K loop of homology model PmSTS and substrate analog from GaDCS-FPF complex structure

Superimpose of ligand FPF from GaDCS (PDB ID: 3G4F in grey) and ligand FPF from NtEAS (PDB ID: 3M01 in orange). The ligand FPF of GaDCS makes a steric clash with the J-K loop of PmSTS (dashed box).

Click here for additional data file.

Table S1 Sesquiterpenes produced by the recombinant PmSTS protein detected using HS-SPME-GC-MS

Click here for additional data file.

We thank Dr. Syarul Nataqain Baharum, Dr. Kamalrul Azlan Azizan, Syahmi Afiq Mustaza, Dr. Tan Cheng Seng, Dr. Teh Aik Hong, Dr. Lee Guan Serm, and Dr. Hong Sok Lai for the technical assistances and scientific discussion. We thank Dr. Paul Dear and Dr. Goh Hoe Han for reading and provides useful comments on the manuscript.

Additional Information and Declarations

Competing Interests

Author Contributions

Data Availability

The authors declare there are no competing interests.

De-Sheng Ker conceived and designed the experiments, performed the experiments, analyzed the data, wrote the paper, prepared figures and/or tables, reviewed drafts of the paper.

Sze Lei Pang conceived and designed the experiments, performed the experiments, analyzed the data, prepared figures and/or tables.

Noor Farhan Othman and Sekar Kumaran conceived and designed the experiments, performed the experiments, analyzed the data.

Ee Fun Tan and Kok Gan Chan contributed reagents/materials/analysis tools.

Thiba Krishnan performed the experiments, prepared figures and/or tables.

Roohaida Othman conceived and designed the experiments, contributed reagents/materials/analysis tools.

Maizom Hassan and Chyan Leong Ng conceived and designed the experiments, analyzed the data, contributed reagents/materials/analysis tools, wrote the paper, prepared figures and/or tables, reviewed drafts of the paper.

The following information was supplied regarding data availability:

The raw data is included in the manuscript in the figures and tables, and in Figs. S2–S5 and Table S1.

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
