# Peer review of "Purification and biochemical characterization of recombinant Persicaria minor β-sesquiphellandrene synthase"

_PeerJ, doi:10.7717/peerj.2961_

## Round 0.1 · original submission · Major Revisions

Please address all issues pointed by the reviewers.

Additional review-level comments by the editor:

The reference for Tantillo, 2011 is missing . Is it Tantillo DJ "Biosynthesis via carbocations: Theoretical studies on terpene formation" Nat. Prod. Rep., 2011, 28, 1035-1053 ?
The reference for Copeland 2000 is also missing.

Is there an online-accessible version of the dissertation of Tan,2012? If so, please include its DOI/handle.

lines 157-162 seem to be mistakenly duplicated from the earlier section

line 222 "Typical plant sesquiterpene synthases with sequences lengths between 500-600 residues were further selected and the proteins that are not producing sesquiterpene were then excluded" please clarify how you detected that a putative sesqiterpene synthase sequence coded for an enzyme which did not produce a sesquiterpene.

line 297: include (computed) self-inhibition Ki used for the continuous line in fig. 3C.

line 367 "It is likely that the side chain of N458 and D459 may still chelate the Mg2+B ion, thereby PmSTS has the catalytic efficiency (kcat/Km) that is comparable to other sesquiterpene synthases (Table 1)." Please provide the rationale behind this speculation: do those sidechains chelate Mg2+ in similar enzymes?

In the sections on multiple sequence alignment and homology model, how dows your analysis differ from that of Su-Fang Ee, Zeti-Azura Mohamed-Hussein, Roohaida Othman, Noor Azmi Shaharuddin, Ismanizan Ismail, and Zamri Zainal, “Functional Characterization of Sesquiterpene Synthase from Polygonum minus,” The Scientific World Journal, vol. 2014, Article ID 840592, 11 pages, 2014. doi:10.1155/2014/840592 ? At first sight, these sections do not appear to provide significant new information. Please clarify, or delete as also suggested by reviewer 3.

Reviewer 1 ·

Basic reporting

The manuscript by Ker et al. described the expression, purification, and characterization of a plant sesquiterpene cyclase PmSTS, which is a class I cyclase that converts farnesyl diphosphate (FPP) to β-sesquiphellandrene. By reporting the protocol, this paper not only addressed the previous inconsistencies of product profiles, but also laid the foundation for future structural and functional studies of such enzyme. This paper is of interest to terpene biosynthesis community, and it can be accepted after the following questions are addressed.

Experimental design

Line 172, authors did not include info on elution buffer composition for affinity column purification.

Line 183, considering the FPP concentration range and Km value, an enzyme concentration of 0.2 microM for determining the Michaelis-Menten parameters seems really high. The authors need to confirm if product conversion is within the linear range for initial rate measurements.

Line 311, the authors should conduct an enzymatic incubation experiment on mutant K266E to support the conclusion that K266E does not affect product profile.

Figure 2A, without glycerol, full-length PmSTS seems to form tetramer in solution; In stark contrast, both PmSTS and PmSTS-delta24 in 15% glycerol are monomeric, regardless of the N-terminal truncation. It is rare that glycerol can substantially change the enzyme oligomerization state, and 15% glycerol is quite high for general enzyme purification and is less physiologically relevant. The authors should try a lower glycerol concentration, such as 5% and 10%, to see how this affects the oligomerization state. In the meantime, a quick dynamic light scattering characterization of peak 5 eluted with and without glycerol can help to determine the homogeneity of purified protein.

Validity of the findings

Line 68, given that there are more than 75,000 terpene molecules identified to date and sesquiterpene is a widely studied subject across fields, 200 sesquiterpenes seem to be a very small pool, the authors should double check the up-to-date number in a natural product database.

The organization of this paper could be modified slightly to make the logic flow: the construct design of PmSTS-delta24 should come before protein expression and purification in the Results section. Another question with this construct design is that, some sequences of plant diterpene cyclases include the plastid sequence, do the first 24 residues constitute the plastid sequence or they are merely disordered and will not be cleaved in mature enzyme PmSTS? Additionally, the authors should test the activity of full-length PmSTS and compare it with that of PmSTS-delta24 to exclude the possibility that deleting the first 24 residues may affect enzymatic activity, so the kinetics measurements obtained from PmSTS-delta24 are relevant.

Line 354, the authors suggest that N458 and D459 may replace the role of last glutamate residue substituted with valine to coordinate to Mg2+B, but it is also likely that a water molecule tethered in place by forming H bonds with surrounding residues completes the hexa-coordination of Mg2+B.

Line 383-386, GaDCS and NtEAS have essentially identical (1,10) cyclization patterns, the authors need to further explain why ligand binding in GaDCS can cause steric clashes.

Additional comments

Line 73, delete ‘into’ after ‘general’, this is a grammatical error.

Figure 4C, C6-C7 double bond configuration of beta-farnesene is incorrect based on the MS pattern, it should be trans instead of cis.

Reviewer 2 ·

Basic reporting

Ker et al. reported the expression and purification of a plant sesquiterpene cyclase from P. minor (PmSTS). Their biochemical and enzymology study confirmed that PmSTS is a β-sesquiphellandrene cyclase, in particularly this cyclase bears an unusual NSE/DTE motif, of which the E residue commonly found in other class I terpene cyclases is substituted by a Val residue in this PmSTS. The results reported here laid a foundation for future structural enzymology and mechanism study. This manuscript is of general interest to natural product (terpene) biosynthesis and terpene cyclase community, and is suitable for publication in PeerJ if the following concerns can be addressed.

There are a lot of grammar errors to which authors should pay attention. For example, in the Background section, 1st line, deleting “that” before synthesized; 3rd line, deleting “for” after encodes; last sentence in this section, insights can not be elucidated. The sentence should be written as “The insights into catalytic active site were further provided by structural analysis guided by multiple sequence alignment”. I strongly suggest the authors should go over the whole manuscript with the help from a native English speaker and correct those grammar errors, even though they do not affect the results and conclusions

Experimental design

The experiment overall is well designed. Only minor questions/suggestions are raised:
1) Have the authors tried to lower the concentration of IPTG during induction? For example, 0.1 mM IPTG could make a huge difference in terms of protein yield.
2) The enzyme concentration is very high relative to substrate concentration. I am afraid the kinetics is not under steady-state when substrate concentration is below 2 µM. Can the authors justify whether their initial rate is still linear?
3) Is the disordered N-terminal peptide playing a signaling role to target the protein into specific organelles, such as plastids? There are a lot of examples like these in plant terpene cyclase, such as taxadiene synthase, epi-aristolochene synthase. Could the authors clarify the potential physiological relevant role of this region? Is it involved in enzymatic function, or just merely a signal peptide?

Validity of the findings

Line 354. The authors propose that both N458 and D459 chelate the Mg2+ ion, which is very unlikely because N458 and D459 are from an alpha-helix in which two adjacent residues are not oriented towards the same direction. And this is why NSE/DTE motifs are residues i, i+4, and i+8 from an bent alpha-helix so that they sit at the same side. I suggest that the authors should use “or” instead of “and”, and if possible, the authors should use site-directed mutagenesis to investigate this unusual and quite interesting motif.

Figure 7. There should be two chiral centers at beta-sesquiphellandrene but the authors did not show or specify them. Besides, the authors use GC-MS to identify products and rely solely on the MS/MS pattern comparison with library. It is necessary but not sufficient to identify terpene products by only MS/MS. The authors should also report the retention index from GC-MS and compare that with the standards in the library.

·

Basic reporting

Basic reporting is generally good. What did confuse me is the lack of the mentioning of the basionym of the plant species used, namely Polygonum minus, which was used in previous work on this gene. The paragraphs on homology modelling and alternative metal binding are not logic in the context of the rest of the paper. They would have been if new data on crystallographic analysis of the purified protein would have been reported. Since this is not the case, I would like to suggest to leave these out, since there is no new experimental confirmation for either present. As for the purification of the enzyme, it is well done, but does not constitute a real breakthrough or a sufficiently novel approach. An important reference on chemical composition in Polygonum/Persicaria species is missing: Prota et al, Phytochemistry 2014.

Experimental design

The experimental design is sound and meets general criteria. The part on homology modelling does not realy provide new insights into the nature of this protein. The same is true, to my opinion, for the motif analysis. This would have been different when crystal data for the protein would have been obtained.

Validity of the findings

Findings are valid. As a general rule, for product identification, I would prefer to have (as a minimum) a Kovats index in combination with MS spectra for comparison with the genuine product. Identification by Nist library hit is unsufficient in my view. But this has been done in previous work, which should have been mentioned in the manuscript. Illustrations are of sufficient quality and clearly presented. Protein purification is well done and documented properly. Enzyme characterization is good, nice to see the substrate inhibition propely taken into account.

---

## Round 0.2 · accepted · Accept

Thank you for addressing all of our reviewer's concerns and for the additional detail provided. The references to Tantillo 2011 and Copeland 2000 are still missing, but you can address that with PeerJ's production staff.